# Frequent Alarm Pattern Mining of Industrial Alarm Flood Sequences by an Improved PrefixSpan Algorithm

Songbai Yang [1,2], Tianxing Zhang [2,3], Yingchun Zhai [1,2], Kaifa Wang [1,2], Guoxi Zhao [4], Yuanfei Tu [2,5,*] and Li Cheng [2,5]

1 Petrochina Tarim Petrochemical Co., Ltd., Korla 841000, China
2 Control Engineering Centre of Nanjing Tech University, Nanjing 210037, China
3 Kunlun Digital Intelligence Technology Co., Ltd., Beijing 100007, China
4 Hexagon's Asset Lifecycle Intelligence, Beijing 100026, China
5 College of Electronic Engineering and Control Science, Nanjing Tech University, Nanjing 210037, China
* Correspondence: yuanfeitu@163.com

**Abstract:** Alarm systems are essential to the process safety and efficiency of complex industrial facilities. However, with the increasing size of plants and the growing complexity of industrial processes, alarm flooding is becoming a serious problem and posing challenges to alarm systems. Extracting alarm patterns from an alarm flood database can assist with an alarm root cause analysis, decision support, and the configuration of an alarm suppression model. However, due to the large size of the alarm database and the problem of sequence ambiguity in the alarm sequence, existing algorithms suffer from excessive computational overhead, incomplete alarm patterns, and redundant outputs. In order to solve these problems, we propose an alarm pattern extraction method based on the improved PrefixSpan algorithm. Firstly, a priority-based pre-matching strategy is proposed to cluster similar sequences in advance. Secondly, we improved PrefixSpan by considering timestamps to tolerate short-term order ambiguity in alarm flood sequences. Thirdly, an alarm pattern compression method is proposed for the further distillation of pattern information in order to output representative alarm patterns. Finally, we evaluated the effectiveness and applicability of the proposed method by using an alarm flood database from a real diesel hydrogenation unit.

**Keywords:** alarm management; industrial alarm systems; alarm flood; PrefixSpan algorithm; sequential pattern recognition

## 1. Introduction

In order to ensure the safety of industrial production, alarm systems are essential to guarantee the safety and efficiency of operations. Alarms are audible or visual signals that alert operators to equipment failures, process deviations, and other abnormalities, thus preventing equipment damage or even production accidents. With the wide use of industrial control systems (ICSs), on the one hand, the cost of designing and configuring alarms has been reduced; on the other hand, the high degree of correlation and complexity between devices makes it possible for a single point of failure to lead to a failure in a related area or even failures in the whole plant, known as cascaded faults. At the same time, unreasonable alarm thresholds and the low performance of alarm management systems in ICSs pose challenges to the efficient operation of alarm systems [1,2], where alarm flooding is the most common and serious problem during the operation of industrial installations. According to EEMUA and ISA 18.2 standards, operators should not receive more than 6 alarms per hour, and alarm flooding is defined as more than 10 alarms per operator per 10 min [3,4]. During alarm floods, operators are unable to identify critical information from numerous alarms in a timely manner, resulting in the lack of effective actions to address critical exceptions, which affects product quality and increases production costs and poses a significant risk to process safety as well as personnel safety. For instance, 275 different

alarms occurred in the 10.7 min prior to the 2005 hydrocarbon plant explosion at a Texas refinery in the United States [5,6]. The operators failed to detect an abnormality in the hydrocarbon fractionation level in its isomerization unit in time, leading to an explosion after the gas-phase component was discharged from the vent stack. Numerous industrial standards and accident analyses have shown that a scientific and reasonable alarm system is important to ensure the safety of industrial processes, to enhance production efficiency, and to guarantee the safety of employees.

Alarm floods have become a common phenomenon in the process industry and pose challenges for alarm systems. To date, extensive studies have been carried out to optimize alarm systems so as to alleviate the effects of alarm floods caused by chattering alarms or reduce the number of these alarm floods. Tulsya et al. [7] designed a delayed alarm strategy for the desired and worst conditions based on minimizing the missed alarm rate and false alarm rate to ensure robustness to non-smooth industrial processes. Wang et al. [8] proposed a method to design a dead bandwidth to suppress the number of chattering alarms and mitigate disturbances to the alarm system. Cheng et al. [9] designed an optimal alarm filter to achieve the best alarm accuracy in the case of the given normal and abnormal statistical distributions.

In the process industry, the switching of some operating states and the propagation of cascaded faults usually generate related alarms. As part of alarm rationalization, alarm flood analysis has also attracted extensive research attention and has become a major branch in handling alarm floods. To date, some modified sequence alignment algorithms have been proposed for the pairwise matching of alarm flood sequences. In Ref. [10], the similarity index between paired sequences was calculated using an improved Smith–Waterman algorithm (SWA) and clustered similar alarm sequences based on the similarity scores. Lai et al. [11] proposed an improved basic local alignment search tool (BLAST) by combining the alarm priority information and timestamp. Simulation experiments showed that the improved BLAST has a smaller computational overhead compared to the modified SWA in [10]. In Ref. [12], a weighted sequential similarity approach is proposed to extract alarm sequence templates for given faults. Based on the extracted fault templates, an improved Needleman–Wunsch algorithm is proposed to isolate alarms caused by identified alarm patterns [13].

Alarm flood analysis extracts alarm sequences from alarm logs and identifies alarm patterns based on the similarity indexes or the frequency of alarm occurrences. The extracted alarm patterns can be used in a root cause analysis [14], alarm display and alarm response improvement [15], or some advanced methods mentioned in the EEMUA, such as fault prediction, online alarm suppression, and so forth. However, extracting patterns from an alarm database with thousands of alarms is time-consuming and requires fairly accurate process knowledge [16]. Fortunately, it is common that some events or abnormalities that occur frequently leave a trace in the A&E log. If such a repeated series of alarms can be detected from historical data, it can help to extract alarm patterns.

Although alarm flood analyses with sequence alignment algorithms were implemented in Refs. [12,13], these methods focus more on clustering similar alarm flood sequences rather than detecting frequent alarm patterns. The Apriori algorithm and its variants are major methods for extracting frequent patterns [17]. However, these algorithms need to construct a large number of candidates as well as frequently scan the database to detect patterns, which results in unaffordable computational overhead when applied to industrial alarm databases. In order to reduce the computational cost, Zhou et al. [18] proposed a modified CloFast algorithm to extract compact alarm patterns in industrial alarm floods.

PrefixSpan (Prefix-Projected Pattern Growth) is an efficient algorithm that generates smaller-sized item databases and provides faster computation [19]. Niyazamand et al. [20] proposed a modified PrefixSpan algorithm (M_PrefixSpan) to extract frequent patterns in alarm flood sequences. However, M_PrefixSpan is based on the premise that alarm sequences are sequential, and such a premise is difficult to satisfy in real industrial processes:

related alarms occur almost simultaneously and in an uncertain order. When several alarm flood sequences occur with order ambiguities, M_PrefixSpan will sequentially extract alarms as prefixes for expansion. As a result, the frequency of some alarms may not meet the minimum support threshold, making it possible for critical alarms to be neglected, affecting the usability of the extracted patterns. Wang et al. [21] reduced the computational cost of the PrefixSpan algorithm by applying an incremental mining strategy. However, it still fails to solve the problem of the sequence ambiguity of alarm sequences; at the same time, the existing algorithm outputs a large number of redundant alarm patterns, which makes it difficult for users to find representative alarm patterns.

Motivated by the problems described above, a compressed alarm pattern mining method based on the PrefixSpan algorithm (CAPM_PrefixSpan) is proposed to further facilitate the root cause analysis of the alarm flood.

The main contributions of this paper are as follows:

1. We propose a pre-matching mechanism based on the similarity scores of pairwise alarm sequences, which can effectively reduce the computational cost when dealing with numerous alarm data.
2. We modified the method of constructing the projection database in the PrefixSpan algorithm, which can help the algorithm avoid the problem of incomplete patterns due to sequence order ambiguity when mining frequent alarm patterns.
3. We propose a compression method to merge similar extracted alarm patterns so as to cluster and compress frequent alarm patterns into a compact alarm sequence, which prevents the output of cumbersome alarm patterns.

The rest of this paper is organized as follows. Section 2 introduces the preliminaries of alarm systems and the PrefixSpan algorithm. Section 3 presents the proposed CAPM_PrefixSpan algorithm. The effectiveness of CAPM_PrefixSpan is verified based on an industrial case in Section 4. Finally, the conclusion is given in Section 5.

## 2. Preliminaries and Problem Description

This section presents the problem of extracting alarm flood patterns from historical alarm data (A&E log) and describes the relevant definitions and algorithms.

### 2.1. Alarms and Alarm Floods

Alarms are generated when process variables exceed their predetermined thresholds and are stored as a set of structured texts in the Alarm and Event Log (A&E log). As shown in Table 1, an alarm contains many attributes, typically including a tag name, an alarm identifier, time information, and an alarm priority [22]. The tag name is the label corresponding to the alarm, and the alarm identifier denotes the alarm type; e.g., "PVHI" (process variable high) and "PVLO" (process variable low) indicate that an analog variable exceeds the high limit or the low limit of the threshold, respectively. The time label records the time that the alarm happens; the alarm priority indicates the importance of the alarm and is usually determined based on factors such as the consequences of ignoring the alarm and the maximum time allowed to deal with the alarm.

Therefore, in this paper, we represent an alarm with a tuple containing three attributes:

$$x_i^j = (e_{x_i^j}, t_{x_i^j}, p_{x_i^j}) \tag{1}$$

where $e_{x_i^j}$ is the alarm label, which is a combination of the tag name and the identifier. $t_{x_i^j}$ and $p_{x_i^j}$ are the time label and the priority of the alarm, respectively. As a result, these three attributes can define an arbitrary unique alarm $x_i^j$ in any alarm sequence $X_j$ in the A&E log.

Based on the ANSI/ISA 18.2 definition of an alarm flood (more than ten alarms per operator per ten minutes), an alarm flood sequence can be expressed as Equation (2) shows.

$$X_j = [x_1^j, x_2^j, \ldots, x_{|X_j|}^j], X_j \in \mathbb{D} \tag{2}$$

where $|X_j|$ is the length of the alarm flood sequence and satisfies $|X_j| \geq 10$ according to the definition. $\mathbb{D}$ is the alarm flood database, which is a collection of all the alarm flood sequences extracted from the A&E log.

**Table 1.** An example of alarms in industrial plants.

| Time Label | Tag Name | Identifier | Alarm Label | Priority |
|---|---|---|---|---|
| 2 May 2021 22:36:01 | PI251 | PVLO | PI251.PVLO | 3 |
| 2 May 2021 22:36:01 | PI104 | PVHI | PI104.PVHI | 3 |
| 2 May 2021 22:37:33 | PI603B | PVLO | PI603B.PVLO | 3 |
| 2 May 2021 22:38:21 | PI604 | PVLO | PI604.PVLO | 3 |
| 2 May 2021 22:38:25 | PI603B | PVLO | PI603B.PVLO | 3 |
| 2 May 2021 22:38:27 | PI604 | PVLO | PI604.PVLO | 3 |
| 2 May 2021 22:38:28 | FI301 | PVLO | FI301.PVLO | 3 |
| 2 May 2021 22:38:29 | FI301 | PVLO | FI301.PVLO | 3 |
| 2 May 2021 22:38:31 | FI301 | PVLO | FI301.PVLO | 3 |
| 2 May 2021 22:38:31 | PI004 | PVHI | PI004.PVHI | 3 |
| 2 May 2021 22:39:40 | FI703 | PVLO | FI703.PVLO | 3 |
| 2 May 2021 22:40:16 | PI004 | PVHI | PI004.PVHI | 3 |

### 2.2. Chattering Alarms

Chattering alarms are large and single alarm messages due to one alarm variable fluctuating around the alarm threshold over a short period of time. Due to the prevalence of noise and unreasonable alarm designs, chattering alarms are very common in the process industry and account for more than 80% of the total number of alarms [23]. Chattering alarms are unable to convey interdependent pattern information between alarms and interfere with the extraction of frequent alarm patterns. Therefore, it is important to remove chattering alarms during the data pre-processing phase.

Clustering identical alarms into a single event can eliminate the influence of chattering alarms. We adopt a predefined time window $T_W$ to eliminate chattering alarms: when an alarm is generated, alarms with the same alarm label and the same alarm identifier for the subsequent duration of the alarm are ignored. After this processing, this method ensures that two identical alarms are separated by at least $T_W$ (s).

Figure 1a shows 135 alarms for diesel flow in the atmospheric and vacuum distillation units of the refinery, and Figure 1b shows the processed alarms by setting the time window $T_W$ = 120 s. Only four alarms are preserved for further alarm pattern analysis.

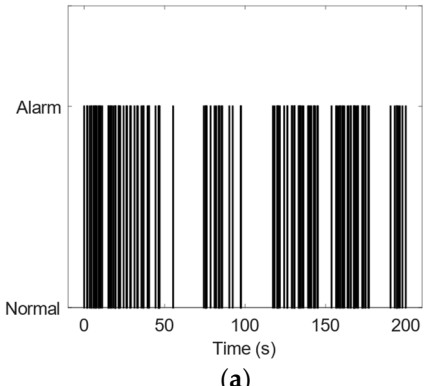
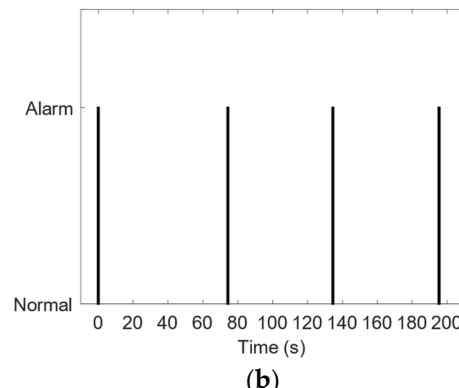

(a)　(b)

**Figure 1.** Alarm for tag "TI203.PVHI". (**a**) Alarm "TI203.PVHI" before removing chattering alarms. (**b**) Alarm "TI203.PVHI" after removing chattering alarms.

### 2.3. Mining Frequent Alarm Patterns with PrefixSpan

PrefixSpan is a variant of the FreeSpan algorithm, which continuously generates and mines smaller projection databases by recursive mining until all items are lower than the

support threshold. In Ref. [21], a modified PrefixSpan (M-PrefixSpan) is proposed for mining frequent alarm patterns. The relevant definitions of M_PrefixSpan are as follows:

Item: Each alarm label in the alarm flood sequence database. For example, the alarm label "PI251.PVLO" in Table 1.

Item frequency (marked as $\xi(\bullet)$): The total number of alarm sequences in $\mathbb{D}$, which contains at least one alarm label.

Support threshold (marked as $Sup_{min}$): The minimum frequency of an item to be considered a candidate as a frequent item.

Prefix and suffix: Consider three alarm sequences $\alpha = \left\{ S_m^1, S_m^2, \ldots, S_m^p \right\}$, $\beta = \left\{ S_n^1, S_n^2, \ldots, S_n^q \right\}$ and $\gamma = \left\{ S_m^{q+1}, S_m^{q+2}, \ldots, S_m^p \right\}$ with $q \leq p$. Alarm sequence $\beta$ is called a prefix of $\alpha$ if $S_n^i = S_m^i, \forall i \leq q$. The remaining sequence $\gamma$ is called a suffix of $\alpha$ with regards to prefix $\beta$.

Projection database: The collection of suffixes for a given prefix in the alarm sequence database.

For instance, Table 2 shows several alarm sequences from a steam generator in a diesel hydrogenation plant. By setting $Sup_{min} = 2$, the steps of mining frequent alarm patterns with PrefixSpan are as follows:

**Table 2.** Alarm sequences for a steam generator in a diesel hydrogenation plant.

| Alarm Sequence #1 | | | Alarm Sequence #2 | | | Alarm Sequence #3 | | |
|---|---|---|---|---|---|---|---|---|
| Time | Alarm Label | Priority | Time | Alarm Label | Priority | Time | Alarm Label | Priority |
| 2021/6/15 5:01:40 | FI702.PVHI | 2 | 2021/6/22 17:43:17 | FI702.PVHI | 2 | 2021/6/24 13:23:12 | FI702.PVHI | 2 |
| 2021/6/15 5:02:08 | LI701.PVHI | 3 | 2021/6/22 17:43:50 | LI701.PVHI | 3 | 2021/6/24 13:23:48 | LI701.PVHI | 3 |
| 2021/6/15 5:02:38 | TI702.PVLO | 3 | 2021/6/22 17:44:14 | TI701.PVLO | 3 | 2021/6/24 13:24:12 | TI701.PVLO | 3 |
| 2021/6/15 5:02:45 | TI701.PVLO | 3 | 2021/6/22 17:44:22 | TI702.PVLO | 3 | 2021/6/24 13:24:26 | TI702.PVLO | 3 |
| 2021/6/15 5:02:52 | TI311.PVLO | 3 | 2021/6/22 17:44:31 | TI407.PVLO | 3 | 2021/6/24 13:24:34 | TI407.PVLO | 3 |
| 2021/6/15 5:03:08 | TI407.PVLO | 3 | 2021/6/22 17:44:50 | PI125.PVHI | 3 | 2021/6/24 13:24:42 | PI125.PVHI | 3 |

Step 1: Scan the database and determine the frequency of each item (alarm) and its frequency: "FI702.PVHI-3", "LI701.PVHI-3", "TI702.PVLO-3", "TI701.PVLO-3", "TI311.PVLO-2", "TI407.PVLO-3", and "PI125.PVHI-1". Since "PI125.PVHI" does not meet the support threshold $Sup_{min}$, "PI125.PVHI" is excluded. Here, we use "FI702.PVHI" as an example of a prefix to expand its frequency pattern.

Step 2: Create a projection database with each prefix.

Step 3: Determine the frequencies of all suffixes associated with the prefix. The frequency of the suffix with regard to the prefix "FI702.PVHI" is shown in Table 3.

**Table 3.** Frequencies of suffixes with regard to the prefix "FI702.PVHI".

| Item | Frequency of the Suffix |
|---|---|
| LI701.PVHI | 3 |
| TI702.PVLO | 3 |
| TI701.PVLO | 3 |
| TI311.PVLO | 2 |
| TI407.PVLO | 3 |

Step 4: The frequencies of all items in Table 3 are greater than the support threshold $Sup_{min}$. The new prefixes are updated to {FI702.PVHI, LI701.PVHI}, FI702.PVHI, TI701.PVLO}, {FI702.PVHI, TI311.PVLO}, and {FI702.PVHI, TI407.PVLO}.

Step 5: Repeat Step 2 to Step 5 until the support values of all items in the projection database are lower than the threshold $Sup_{min}$.

Step 6: Remove alarm patterns that are subsets of other patterns. Finally, the frequent alarm patterns are {FI702.PVHI, LI701.PVHI, TI702.PVLO, TI407.PVLO}, {FI702.PVHI,

LI701.PVHI, TI701.PVLO, TI407.PVLO}, {FI702.PVHI, LI701.PVHI, TI701.PVLO, TI702.PVLO}, and {FI702.PVHI, LI701.PVHI, TI311.PVLO, TI407.PVLO}.

### 2.4. Problem Description

In summary, the major aim of this paper is to extract frequent alarm patterns from the A&E logs of industrial alarm systems. As shown in Figure 2, after removing chattering alarms, the calculation for extracting alarm patterns from the alarm database $\mathbb{D}$ is conducted in the following three steps:

1.  The priority-based pre-matching strategy is used to cluster similar alarm flood sequences so as to reduce the computational overhead.
2.  Closed frequent alarm patterns are discovered to extract typical alarm patterns.
3.  The alarm pattern is compressed to reduce the impact of cumbersome frequent alarm patterns.

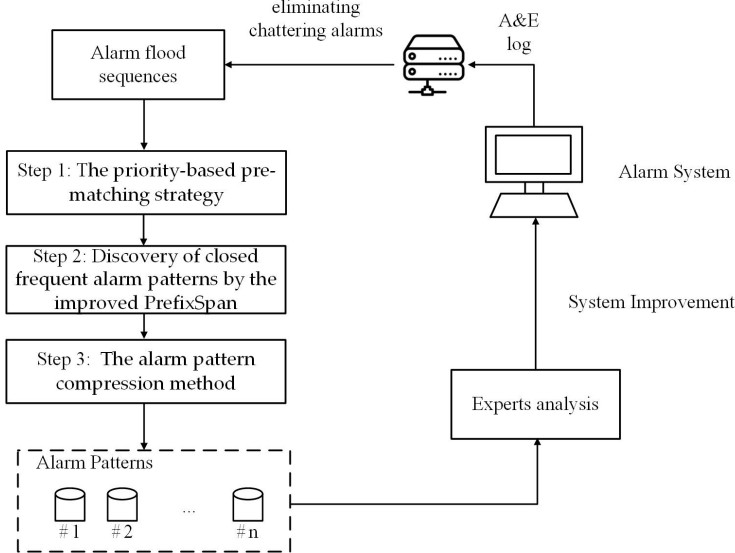

**Figure 2.** Framework of the proposed method for mining alarm patterns.

The specific calculations and processes for the above steps are described in the following section.

## 3. Proposed Methods

In this section, we detail the three steps of CAPM_PrefixSpan for mining frequent alarm patterns, including the priority-based pre-matching strategy, the discovery of closed frequent alarm patterns, and the alarm pattern compression method.

### 3.1. The Priority-Based Pre-Matching Strategy

The pre-matching strategy utilizes the similarity index of alarm sequences or alarm attributes of alarm sequences to cluster similar alarm sequences and thus exclude irrelevant ones. As one of the important attributes of alarms, the alarm priority indicates the importance of the alarm. As shown in Table 4, there are usually three or four alarm priorities established in industrial alarm systems. According to ISA 18.2, for alarm systems with three levels of alarm priorities, the recommended percentage for each alarm priority from "Low" to "Emergency" should be 80%, 15%, and 5%, respectively. High-priority alarms have a smaller percentage but indicate severe abnormal conditions; on the contrary, lower priorities are typically used to configure most of the less severe alarms. Thus, it is reasonable to cluster alarm flood sequences based on the co-occurrence of alarms with higher priorities.

**Table 4.** Typical alarm priorities of industrial alarm systems.

| Type 1 | Type 2 | Type 3 | Corresponding Priority |
|---|---|---|---|
| Emergency | Emergency | Critical | 1 |
| High | High | Warning | 2 |
| Medium | Low | Advisory | 3 |
| Low | | | 4 |

In order to cluster similar alarm flood sequences in a given database, a binary matrix $S$ is established with each element calculated by Equation (3):

$$S_{i,j} = \begin{cases} 1, & s_{(X_i, X_j)} \geq \mu_{th} \\ 0, & \text{otherwise} \end{cases} \tag{3}$$

where $S_{i,j}$ represents the element in the i-th row and j-th column of matrix $S$, $s_{(X_i, X_j)}$ is the similarity index between the i-th and j-th alarm sequences, and $\mu_{th}$ is the threshold for matching similar sequences.

$$s_{(X_i, X_j)} = \begin{cases} \sum_{n=1}^{m} \psi(\mathbb{C}_{i,j}^n) & \mathbb{C}_{i,j} \neq \varnothing \\ 0 & \mathbb{C}_{i,j} = \varnothing \end{cases} \tag{4}$$

where $\mathbb{C}_{i,j} = \{e_x | x = (e_x, t_x, p_x) \in X_i \cap X_j\}$ is the collection of all identical alarm labels between the i-th and j-th alarm sequences, $m$ represents the length of set $\mathbb{C}_{i,j}$, and $\mathbb{C}_{i,j}^n$ denotes the n-th alarm label in set $\mathbb{C}_{i,j}$. The function $\psi(\bullet)$ calculates the score based on the priority levels and can be expressed as Equation (5):

$$\psi(e_x) = 2 + 1.5(p_{\max} - p_x) \tag{5}$$

where $p_x$ is the priority level of alarm $x$, $p_{\max}$ is the maximum priority level, and $\beta$ is a positive constant. Therefore, $\psi(e_x)$ increases as the alarm priority increases. For instance, given an alarm system with three priority levels, "High", "Medium", and "Low", the scores $\psi(e_x)$ are 2, 3.5, and 5, respectively.

Once the binary matrix $S$ is finished, alarm flood sequences with $S_{i,j} = 1$ are clustered to segment the alarm flood database $\mathbb{D}$, denoted as $\Re = \left\{ \mathbb{R}_1, \mathbb{R}_2, \ldots, \mathbb{R}_{|\mathbb{R}|} \right\}$.

In each group $\mathbb{R}_k = \left\{ X_1^k, X_2^k, \ldots, X_{|\mathbb{R}_k|}^k \right\}, k \in [1, |\mathbb{R}|]$, any two sequences $X_a^k$ and $X_b^k$ share at least one instance of identical alarm labels. As a result, the algorithm avoids distractions from irrelevant alarm sequences. In the next step, frequent alarm patterns are discovered recursively for each group of alarm sequences $\left\{ \mathbb{R}_1, \mathbb{R}_2, \ldots, \mathbb{R}_{|\mathbb{R}|} \right\}$ by incorporating temporal information.

*3.2. Discovering Closed Frequent Alarm Patterns*

**Definition 1:** *Alarm pattern P is a frequent alarm pattern if the frequency $\xi(P) \geq Sup_{\min}$. Notice that if P is frequent, all subsets of P are also frequent.*

**Definition 2:** *Alarm pattern P is a closed alarm pattern iff:*

(1)    Alarm pattern $P$ is frequent.

(2)    There is no frequent pattern $P'$, such that $P \subset P'$ and $\xi(P) = \xi(P')$.

The PrefixSpan algorithm builds the projection database by counting the frequency of each suffix corresponding to the prefix. In order to reduce interference from the order

ambiguity problem in alarm flood sequences, we propose an improved projection database construction method by introducing the temporal information of the alarms.

Without loss of generality, for the arbitrary prefix $v$ with the frequency $m$, assume that the suffix $v$ has $n$ alarm sequences containing the alarm tag $e_{x_i}$ $(m \geq n)$. Let $t_v^{(x_i)}$ denote the time information of alarm $x_i$ corresponding to prefix $v$, where $t_v^{(x_i)}$ is an $n \times 1$ column vector. Let $t^{(v)}$ denote the final time of $m$ alarm sequences with respect to prefix $v$. The time distance matrix $\Delta T_{v,x_i}$ can be calculated by the following equation.

$$\Delta T_{v,x_i} = [\Delta t]_{n \times m} = t_v^{(x_i)} \cdot I_{1 \times m} - I_{n \times 1} \cdot t^{(v)\mathrm{T}} \tag{6}$$

where $I_{1 \times m}$ and $I_{1 \times n}$ are the unit vector. The time span $T_s = [t_b, t_f]$ is used to truncate the time distance matrix $\Delta T_{v,x_i}$, where $t_b$ is a negative constant, so as to reduce the impact of sequence order ambiguity; $t_f$ is the time window and a positive constant (usually set to 100 s), which is used to determine the causal relationship between the alarms. To truncate the time distance matrix $\Delta T_{v,x_i}$ by the time span $T_s$ according to Equation (7), we have $\nabla T_{v,x_i} = [\nabla t]_{n \times m}$:

$$\nabla t_{ij} = \begin{cases} 1, & \text{if } \Delta t_{ij} \in T_s \\ 0, & \text{otherwise} \end{cases} \tag{7}$$

where $\Delta t_{ij}$ is an element in the time distance matrix. $\Delta t_{ij} \in (0, t_f]$ indicates that alarm $x_i$ occurs after prefix $v$ within $t_f$ seconds; $\Delta t_{ij} \in [t_b, 0]$ indicates that alarm $x_i$ occurs within $t_s$ seconds before prefix $v$. Therefore, the frequency of the alarm tag can be calculated as:

$$\zeta(e_{x_i} \to v) = rank(\nabla T_{v,x_i}) \tag{8}$$

where $\zeta(e_{x_i} \to v)$ represents the total frequency of alarm tag $e_{x_i}$ following prefix $v$. Then, alarm tag $e_{x_i}$ and corresponding frequency $\zeta(e_{x_i} \to v)$ will be recorded in the projection database $\mathbb{B}_{w.r.t\ v}$. The Pseudocode for constructing the projection database is summarized in Algorithm 1.

If $\zeta(e_{x_i} \to v) \geq Sup_{min}$, alarm tag $e_{x_i}$ is integrated with the current corresponding prefix $v$ to construct a new prefix, $v'$. Further, the average time interval between $e_{x_i}$ and $v$ can be calculated as $\tau(e_{x_i} \to v)$ and will also be recorded. The prefix extension is recursively performed until no more frequent alarm items can be found in the projection database. The major codes for building the projection database are summarized in Algorithm 1. Finally, a filter is adopted to remove all the subsets of frequent alarm patterns, rendering the final output of the CAPM_PrefixSpan a closed frequent alarm pattern. The main codes for discovering closed frequent alarm patterns are shown in Algorithm 2.

---

**Algorithm 1** Major codes for building projection database

---

1 Input: $v$, $T_s$, $\mathbb{R}_k$
2 Output: $\mathbb{B}_{w.r.t\ v}$
3 $\mathbb{X}$=The set of all alarm tags in the suffix with regard to prefix $v$
4 For each alarm tag $e_{x_i}$ in $\mathbb{X}$.
5 　$t^v$= timestamp of $v$ in $\mathbb{R}_k$
6 　$t_v^{(x_i)}$ = timestamp of $x_i$ in $\mathbb{R}_k$
7 　Calculate time distance matrix $\Delta T_{v,x_i}$ according to Equation (6)
8 　$\nabla T_{v,x_i}$= Truncate $\Delta T_{v,x_i}$ by time span $T_s$ according to Equation (7)
9 　$rank(\nabla T_{v,x_i})$ = the frequency of alarm tag $e_{x_i}$ with respect to $v$ in $\mathbb{R}_k$
10 　Add $\{e_{x_i}:rank(\nabla T_{v,x_i})\}$ into $\mathbb{B}_{w.r.t\ v}$
11 End For

---

---

**Algorithm 2** Major codes for discovering closed frequent alarm patterns in $\Re$

---

1 Input: $\Re$, $T_s$, $Sup_{min}$
2 Output: $\mathbb{P}$
3 For each $\mathbb{R}_k$ in $\Re$
4 Scan all alarm items in $\mathbb{R}_k$.
5 Remove items with frequencies lower than $Sup_{min}$.
6 $v_{current}$ = the remaining alarm items in $\mathbb{R}_k$.
7
8 For Each $v_i$ in $v_{current}$
9   $\mathbb{B}_{w.r.t\ v_i}$ = Projection database calculated by Algorithm 1
10     For each alarm tag $e_{x_n}$ in $\mathbb{B}_{w.r.t\ v_i}$
11     If $\zeta(e_{x_n} \to v_i) \geq Sup_{min}$
12       Update $v_{next}$ by Assembling $e_{x_n}$ with $v_i$
13     End If
14     End For
15 End For
16 While $v_{current} \neq v_{next}$
17 $v_{current} = v_{next}$
18 For each $v_m$ in $v_{current}$
19   $\mathbb{B}_{w.r.t\ v_m}$ = projection database with respect to $v_m$ by Algorithm 1
20     For each $e_{x_n}$ in $\mathbb{B}_{w.r.t\ v_m}$
21     If $\zeta(e_{x_n} \to v_m) \geq Sup_{min}$
22       Update $v_{next}$ by Assembling $e_{x_n}$ with $v_m$
23     End If
24     End For
25 End For
26 End While
27 $P_k$ = remove all subsets of $v_{current}$
28 $\mathbb{P} = \left\{ P_1, P_2 \dots P_{|\mathbb{R}|} \right\}$
29 End For

---

### 3.3. Frequent Alarm Pattern Compression Method

Even if similar alarm flood sequences are clustered by the pre-matching strategy proposed in Section 3.1, the closed frequent alarm patterns extracted from $\mathbb{R}_k$ are still numerous and redundant. Therefore, the extracted frequent alarm patterns should be further compressed into representative alarm patterns.

For the given closed alarm pattern $P_k$ extracted from $\mathbb{R}_k$, a binary matrix $Z_k$ is created by calculating the pairwise similarity index between patterns $P_a^k$ and $P_b^k$:

$$Z_{a,b}^k = \begin{cases} 1, & \text{if } \varphi(\mathrm{P}_a^k, \mathrm{P}_b^k) \geq \gamma \\ 0, & \text{otherwise} \end{cases} \tag{9}$$

where $\gamma$ is the threshold for pattern compression, and $Z_{a,b}^k$ denotes an element of the i-the column and j-th row in matrix $Z$. The function $\varphi(\mathrm{P}_a^k, \mathrm{P}_b^k)$ calculates the similarity index between $P_a^k$ and $P_b^k$. Since the closed alarm patterns extracted from the same collection $\mathbb{R}_k$ usually share identical alarm tags, the similarity index $\varphi(\mathrm{P}_a^k, \mathrm{P}_b^k)$ is calculated based on the optimal alignment of the optimal segment pairs of two alarm patterns. As a result, the Smith–Waterman algorithm is utilized to calculate $\varphi(\mathrm{P}_a^k, \mathrm{P}_b^k)$ as Equation (10) shows.

$$\varphi\left(P_a^k, \mathrm{P}_b^k\right) = \frac{\max\limits_{1 \leq i \leq p \leq |p_i^k|, 1 \leq j \leq q \leq |p_j^k|} \left( \phi\left(P_{i:m}^a, P_{j:n}^b\right), 0 \right)}{\min\left(\phi\left(P_a^k, P_a^k\right), \phi\left(P_b^k, P_b^k\right)\right)} \tag{10}$$

where $\phi\left(P^a_{i:m}, P^b_{j:n}\right)$ is the similarity index of the segmented pair $\left(P^a_{i:m}, P^b_{j:n}\right)$. In order to find the best local alignment between $P^k_a$ and $P^k_b$, the SW algorithm recursively calculates an index matrix $H$:

$$H_{p+1,q+1} = \max\left\{H_{p,q} + \rho\left(x^a_p, x^b_q\right), H_{p,q+1} + \delta, H_{p+1,q} + \delta, 0\right\} \tag{11}$$

where $H_{p+1,q+1}$ is an element of the matrix, and $\delta$ is the gap penalty. For any $p$ and $q$, $H_{1,q} = 0$ and $H_{p,1} = 0$ since one or both of the segments of $P^k_a$ and $P^k_b$ are empty. $\rho\left(x^a_p, x^b_q\right)$ is the similarity score function, as Equation (12) shows:

$$\rho\left(x^a_p, x^b_q\right) = \begin{cases} 1, & \text{if } e_{x^a_p} = e_{x^b_q} \\ -0.6, & \text{if } e_{x^a_p} \neq e_{x^b_q} \end{cases} \tag{12}$$

Based on Equation (11), matrix H and the similarity index $\varphi(P^k_a, P^k_b)$ can be worked out. Following this, by calculating all alarm patterns in $P_k$, the matrix $Z_k$ can be obtained. By clustering the alarm patterns with $Z_k = 1$, similar closed alarm pattern collections can be recognized for further compression. These collections can be expressed as:

$$N_k = \left\{C^k_1, C^k_2, ..., C^k_{|N_k|}\right\} \tag{13}$$

where $C^k_m = \left\{P^k_1, P^k_2, \ldots, P^k_{|C^k_m|}\right\}, m \in [0, |N_k|]$ is the clustered alarm pattern, and $k = 1, 2, \ldots, |\mathbb{R}|$ is the index of the extracted alarm pattern in $\mathbb{P}$.

Finally, the compressed alarm pattern $Y^k_m$ is distilled from each $C^k_m$ in according to Equation (14):

$$Y^k_m = P^k_1 \oplus P^k_2 \oplus \ldots \oplus P^k_{|C^k_m|} \tag{14}$$

where the operator $\oplus$ indicates the combination of elements in the clustered frequent alarm patterns based on their corresponding average timestamp. The Pseudocode of the frequent alarm pattern compression method is shown in Algorithm 3.

---

**Algorithm 3** Major codes for compressing alarm patterns in $\mathbb{P}$

---

1 Input: $\mathbb{P}, \gamma$
2 Output: $\mathbb{Y}$
3 For each $P_k$ in $\mathbb{P}$
4 For i = 1: Length($P_k$)
5 For j = 1: Length($P_k$)
6 Calculate the index matrix $H$ between $P^k_i$ and $P^k_j$ based on Equation (11)
7 Calculate similarity index $\varphi(P^k_i, P^k_j)$
8 If $\varphi(P^k_i, P^k_j) \geq \gamma$
9 $Z^k_{i,j} = 1$
10 Else:
11 $Z^k_{i,j} = 0$
12 End If
13 End For
14 End For
15 Cluster the closed frequent alarm patterns with $Z_k = 1$
16 $N_k$ = the collection of the clustered alarm patterns
17 $Y^k_m$ = compress the alarm patterns in each $C^k_m \in N_k$ according to Equation (14)
18 Add $Y^k_m$ into $\mathbb{Y}$
19 End For

---

*3.4. Implementation Procedure*

The major steps for mining frequent alarm patterns by CAPM_PrefixSpan are summarized in Algorithm 4, where $\mathbb{D}$ is the alarm flood sequences, $[T_w \ \mu_{th} \ \gamma \ T_s \ Sup_{min}]$ are the predefined parameters, and $\mathbb{Y}$ denotes the set of compressed closed alarm patterns. The detailed steps are as follows:

Step 1. Remove chattering alarms by using the time window $T_w$.

Step 2. Calculate the similarity score of all alarm flood sequences in $\mathbb{D}$ based on Equation (4) and cluster similar alarm sequences according to Equation (3).

Step 3. Extract closed frequent alarm patterns recursively from the set $\Re = \left\{ \mathbb{R}_1, \mathbb{R}_2, \ldots, \mathbb{R}_{|\mathbb{R}|} \right\}$ according to Algorithm 2.

Step 4. Compress the extracted alarm patterns for each collection in $\mathbb{P} = \left\{ P_1, P_2 \ldots P_{|\mathbb{R}|} \right\}$ according to Algorithm 3.

---

**Algorithm 4** Mining closed frequent alarm patterns in $\mathbb{D}$

---

1 Input: $\mathbb{D}$, $T_w$, $\mu_{th}$, $T_s$, $\gamma$, $Sup_{min}$
2 Output: $\mathbb{Y}$
3 Remove chattering alarms in $\mathbb{D}$.
4 Divide $\mathbb{D}$ into $\Re = \left\{ \mathbb{R}_1, \mathbb{R}_2, \ldots, \mathbb{R}_{|\mathbb{R}|} \right\}$ by using the priority-based pre-matching strategy.
5 For each pattern in $\mathbb{R}$ in $\Re$
6 Mining closed frequent alarm patterns $\mathbb{P}$ according to Algorithm 2
7 End for
8 Compress the alarm patterns $\mathbb{P}$ into $\mathbb{Y}$ according to Algorithm 3

---

In the CAPM_PrefixSpan algorithm, several important parameters are involved, including the time window $T_w$, pre-matching threshold $\mu_{th}$, time span $T_s$, compress threshold $\gamma$, and minimum support threshold $Sup_{min}$. For the easier implementation of the algorithm for practitioners, the following guidelines can be considered when selecting the parameters of CAPM_PrefixSpan.

1. In the data processing step, $T_w$ specifies the minimum time interval between two identical alarm tags. By default, $T_w = 100s$ to filter chattering alarms is widely used in practice [20].

2. In the pre-matching stage, $\mu_{th}$ specifies the minimum similarity score for the pre-matching strategy. For the alarm system with three levels of priorities, $\mu_{th} = 5 \times 2 = 10$ is set because ISA 18.2 considers an alarm flood to be over when the alarm rate is less than five alarms in 10 min; in addition, ISA 18.2 suggests that 80 percent of the alarms should be designated "Low" priority alarms, which have a similarity score of 2 according to Equation (5).

3. In the closed frequent alarm pattern discovery stage, $Sup_{min}$ specifies the minimum occurrence frequency for considering an alarm to be a frequent alarm in the analyzed alarm floods. By default, $Sup_{min} = 2$ is set to capture all repeated alarms. $T_s$ specifies the tolerance of order ambiguity in the alarm flood sequences. By default, $T_s = [-10, 100]$ is set as the tolerance of short-term order ambiguity to discover casualty alarms.

4. In the alarm pattern compression stage, $\gamma$ specifies the threshold for merging similar alarm patterns. The value of $\gamma$ can be determined based on user requirements.

## 4. Industrial Case Study

In this section, we intend to evaluate the performance of the proposed CAPM_PrefixSpan algorithm in terms of computational cost and mining alarm patterns based on a real industrial alarm sequence database.

### 4.1. Data Acquisition and Comparative Algorithms

The experimental data were collected from the A&E log of a typical diesel hydrogenation unit at a refinery in Xingjiang Province, China. This facility had a total of 2203 configured alarms for monitoring 503 process variables. The alarm data were extracted from April 2020 to August 2020, and the chattering alarms were removed by setting $T_w = 100s$. Finally, 161 alarm flood sequences were extracted according to the definition in ISA 18.2. Details of the extracted alarm flood sequences are shown in Table 5.

**Table 5.** Details of the extracted alarm flood sequence database.

| Attribute | Description |
|---|---|
| Number of alarm flood sequences | 161 |
| Average length of sequences | 27.2 |
| Length of the longest sequence | 41 |
| Length of the shortest sequence length | 10 |
| Average duration of alarm flood | 863.4 (s) |
| Longest duration of alarm flood | 1357 (s) |
| Shortest duration of alarm flood | 392 (s) |

In order to evaluate the effectiveness of the CAPM_PrefixSpan algorithm, M_PrefixSpan [20] and causality PrefixSpan (C_PrefixSpan) [21] were compared on the same alarm flood database. In Ref. [21], the C_PrefixSpan algorithm utilized the time span $t_f$ to capture causality alarms associated with prefixes. The parameter settings are shown in Table 6.

**Table 6.** Detailed parameters of each algorithm.

| Algorithm | $Sup_{min}$ | $t_f$ | $\mu_{th}$ | $t_b$ | $\gamma$ |
|---|---|---|---|---|---|
| M_PrefixSpan | 2 | - | - | - | - |
| C_PrefixSpan | 2 | 100 | - | - | - |
| CAPM_PrefixSpan | 2 | 100 | 10 | −10 | 0.6 |

### 4.2. Comparison of the Overall Results

Firstly, a total of 29 collections of clustered alarm flood sequences were obtained by using the pre-matching strategy proposed in Section 3.1. Next, frequent alarm pattern mining and alarm pattern compression were recursively performed for each clustered alarm sequence. As a result, we obtained a total of 121 closed alarm patterns and 33 compressed alarm patterns, which are shown in Figure 3.

By dividing the alarm flood database into 29 alarm collections by using the proposed pre-matching strategy, the search space of the algorithm is effectively reduced. In clustered alarm sequence #1 in Figure 2, this collection contains a total of 32 different alarm tags, which means that the maximum number of alarm tags to be considered is reduced to 32. Without the pre-matching process, all 2203 alarm tags in the DCS of the diesel hydrogenation unit would be examined, resulting in a large number of redundant alarm patterns and potentially unaffordable computational overhead.

To further illustrate the importance of the pre-matching method, Figure 4 presents the running times of the different algorithms for different database sizes. Overall, the larger the database, the longer it takes to extract the closed alarm patterns. This is because, as the database contains more alarm tags and alarm sequences, more alarm tags satisfy $Sup_{min}$ in each iteration, making the size of the projected database also increase dramatically. By clustering alarm flood sequences based on the alarm priority and co-occurrence, the number of alarms to be examined is reduced significantly, which greatly reduces the computation time of the algorithm. Furthermore, this also allows CAPM_PrefixSpan to extract alarm patterns based on a low support threshold.

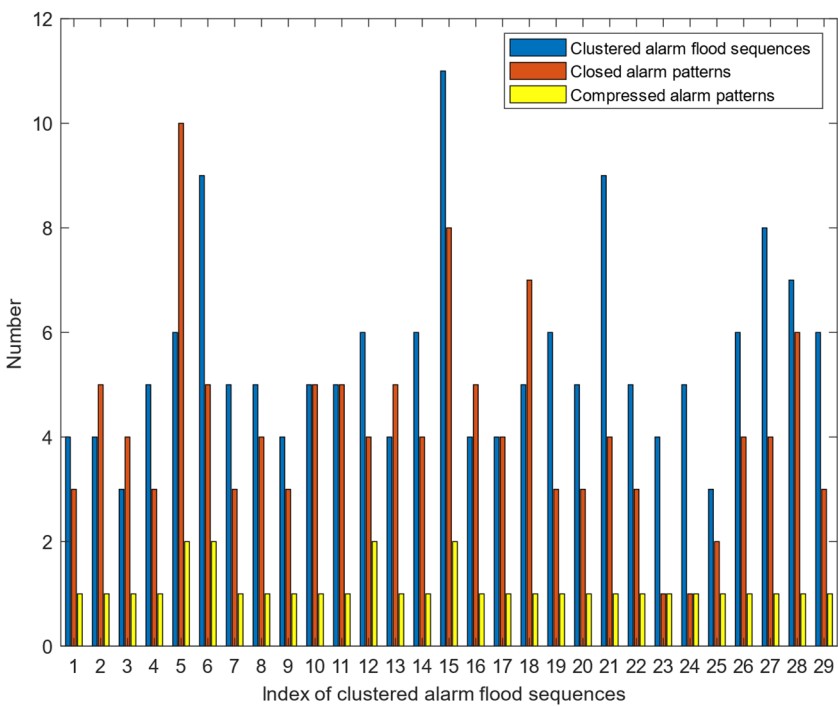

**Figure 3.** Numbers of clustered alarm flood sequences, closed alarm patterns, and compressed alarm patterns from 29 alarm flood groups.

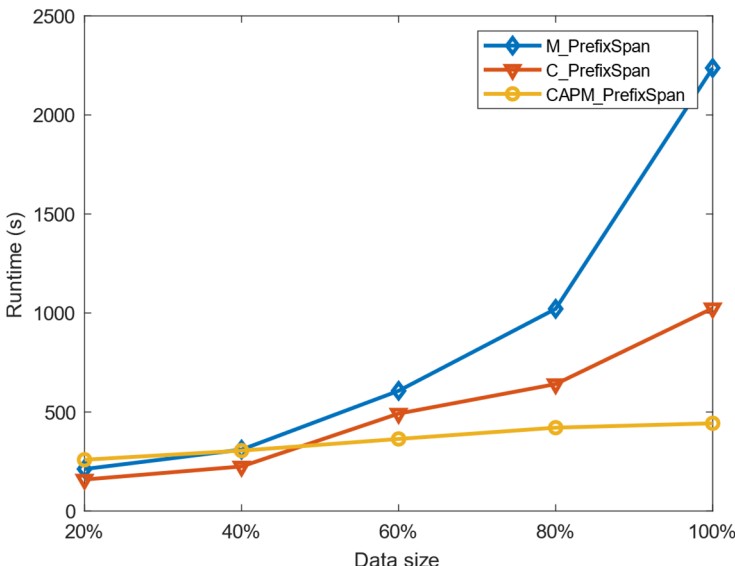

**Figure 4.** Runtime with different sizes of alarm flood sequence databases.

Table 7 shows the number of alarm patterns extracted by the compared algorithms with the full database. CAPM_PrefixSpan extracted the most closed alarm patterns; this is because C_PrefixSpan and M_PrefixSpan are affected by alarm sequence order ambiguity, resulting in the incomplete extraction of alarm patterns. CAPM_PrefixSpan tolerates short-term sequence timing order ambiguity by introducing alarm time information and the time span $T_s$. In addition, it is clear that the alarm pattern compression method significantly reduced pattern redundancy. For example, in alarm flood collection #2, five closed alarm patterns were obtained, and these alarm patterns differ only partially in their alarm tags and are highly similar. The alarm pattern compression method distills similar alarm patterns

into a compressed alarm pattern, which reduces pattern redundancy and in turn helps users to focus on the patterns they are interested in.

**Table 7.** The number of extracted alarm patterns.

|  | M_PrefixSpan | C_PrefixSpan | CAPM_PrefixSpan |
|---|---|---|---|
| Number of closed alarm patterns | 106 | 86 | 121 |
| Number of compressed alarm patterns | / | / | 33 |

*4.3. Comparison of Extracted Alarm Patterns*

For further demonstration, we compared the alarm patterns extracted by the three algorithms from the same collection of alarm sequences.

The alarm sequence comes from the feedstock feed system in the diesel hydrogenation plant. This system feeds the feedstock from the atmospheric depressurization unit to the buffer tank, outputs the feedstock through the booster pump, and removes the fine particles through the filter; it then heats up and feeds it to the buffer tank to provide a stable feed to the downstream unit, which is an important part of the system in the diesel hydrogenation plant.

After the pre-matching process, alarm sequence collection #1 was obtained with a total of four alarm flood sequences. The alarm patterns extracted by M_PrefixSpan, C_PrefixSpan, and CAPM_PrefixSpan from alarm collection #1 are shown in Tables 8–10, respectively. "PDI203" is the filter's differential pressure indicator; "PI106" is the buffer tank pressure indicator; "FI102" and "FI101" are the tank diesel flow indicators; "FI401" is the filter outlet flow indicator; and "FI402" is the atmospheric diesel flow indicator.

**Table 8.** Alarm patterns extracted by M_PrefixSpan.

| Index | Closed Alarm Patterns | Frequency |
|---|---|---|
| 1 | PDI203.PVHI, PI106.PVHI, FI102.PVHI, FI101.PVHI | 2 |
| 2 | PDI203.PVHI, FI101.PVHI, FI102.PVHI | 2 |
| 3 | PDI203.PVHI, PI106.PVHI, FI402.PVHI | 2 |
| 4 | PDI203.PVHI, FI401.PVLO, FI402.PVHI | 2 |
| 5 | PDI203.PVHI, FI401.PVLO, PI106.PVHI, F101.PVHI | 2 |

**Table 9.** Alarm patterns extracted by C_PrefixSpan.

| Index | Closed Alarm Patterns | Frequency |
|---|---|---|
| 1 | PDI203.PVHI, PI106.PVHI, FI102.PVHI | 2 |
| 2 | PDI203.PVHI, FI101.PVHI, FI102.PVHI | 2 |
| 3 | PDI203.PVHI, PI106.PVHI, FI402.PVHI | 2 |
| 4 | PDI203.PVHI, FI401.PVLO, PI106.PVHI | 2 |

**Table 10.** Alarm patterns extracted by CAPM_PrefixSpan.

| Index | Closed Alarm Patterns | Frequency |
|---|---|---|
| 1 | PDI203.PVHI, PI106.PVHI, FI102.PVHI, FI101.PVHI | 3 |
| 2 | PDI203.PVHI, PI106.PVHI, FI101.PVHI, FI102.PVHI | 3 |
| 3 | PDI203.PVHI, FI401.PVLO, FI402.PVHI, PI106.PVHI, FI102.PVHI | 2 |

The process values in collection #1 are highly correlated with each other, but the alarm sequences do not have a fixed order, and the same alarm pattern can produce multiple forms. For example, pattern #1 and pattern #2 in Table 10 are different forms of the same alarm pattern. M_PrefixSpan calculates each frequency separately based on each particular

form, resulting in a reduced frequency for some pattern forms and causing the frequency of that alarm pattern form to not meet the minimum support threshold $Sup_{min}$. As shown in Table 8, it can be found that alarm pattern #1 to pattern #3 are subsets of pattern #1 in Table 10, which indicates that the M_PrefixSpan algorithm fails to extract the complete alarm patterns. Meanwhile, the C_PrefixSpan algorithm also fails to completely extract pattern #1 in Table 8 due to the limited time window and the influence of order ambiguity in alarm flood sequences.

In contrast, the alarm patterns extracted by CAPM_PrefixSpan contained all the patterns in Tables 8 and 9. This indicates that tolerating short-term order ambiguities by setting the time span can effectively improve the mining performance of the alarm patterns.

It is clear to see that pattern #1 and pattern #2 in Table 10 are different forms of the same mode. Therefore, the closed alarm patterns need to be further processed to prevent outputting redundant patterns. Table 11 shows the compressed alarm patterns obtained by CAPM_PrefixSpan. The compressed alarm patterns greatly reduce the pattern redundancy. Further, based on an evaluation by experts with process knowledge, the alarm pattern "PDI203.PVHI" is triggered by high differential pressure due to the backflushing of the oil circuit. After 6.3 s, "FI401.PVLO" was triggered. In addition, the high flow of diesel fuel "FI405.PVHI", "FI102.PVHI", and "FI101.PVHI" caused the high pressure of the tank and triggered alarm "PI106.PVHI" within 12.4 s. Therefore, this alarm mode makes sense because it effectively exposed the fault propagation path in the plant.

**Table 11.** Compressed alarm patterns obtained by CAPM_PrefixSpan.

| Compressed Alarm Patterns Extracted by CAPM_PrefixSpan | | |
|---|---|---|
| Pattern Index | Alarm Tag | Time (s) |
| | PDI203.PVHI | 0 |
| | FI401.PVLO | 6.3 |
| 1 | FI402.PVHI | 7.2 |
| | PI106.PVHI | 12.4 |
| | FI102.PVHI | 16.8 |
| | FI101.PVHI | 17.1 |

## 5. Conclusions

In the process industry, alarm sequences caused by the same propagation path share different forms because of noise and the randomness of detection delays. In order to facilitate alarm pattern extraction as well as improve alarm systems, an alarm pattern extraction method is proposed, which consists of three main stages: the pre-matching strategy based on alarm priority, the improved PrefixSpan algorithm, and the alarm pattern compression method. To verify the effectiveness of the proposed method, an industrial case study was carried out with alarm data from a complex facility of a refinery. The experimental results show that CAPM_PrefixSpan improves the efficiency of alarm pattern recognition by introducing alarm timestamp information and tolerating short-term order ambiguity. In addition, the effectiveness of the compressed alarm patterns was verified by an expert evaluation.

However, alarm pattern mining based on historical data can only extract alarm patterns from abnormalities that have occurred. Furthermore, as the proposed algorithm is based on the number of occurrences of alarm tags in a particular DCS system, the extracted alarm patterns are still not universal across the same processes at different facilities. Therefore, future work will focus on investigating generalized alarm pattern mining methods.

**Author Contributions:** Conceptualization, S.Y., T.Z. and G.Z.; methodology, Y.Z.; validation, K.W., T.Z. and G.Z.; writing—original draft preparation, Y.T. and G.Z.; writing—review and editing, L.C.; funding acquisition, Y.T. All authors have read and agreed to the published version of the manuscript.

**Funding:** This work is supported by the Postgraduate Research & Practice Innovation Program of Jiangsu Province under Grant No. SJCX22_0420.

**Data Availability Statement:** Not applicable.

**Conflicts of Interest:** The authors declare no conflict of interest.

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
