# Peer review of "Frequent Alarm Pattern Mining of Industrial Alarm Flood Sequences by an Improved PrefixSpan Algorithm"

_processes, doi:10.3390/pr11041169_

Round 1

Reviewer 1 Report

The manuscript is well organized and written, with a coherent and logical presentation of the arguments and findings. The results and arguments are readily comprehensible and thoroughly analyzed.

Author Response

Response to Reviewer 1 Comments

Dear reviewer:

Thank you for your decision and constructive comments on my manuscript.

Reviewer #1:

The manuscript is well organized and written, with a coherent and logical presentation of the arguments and findings. The results and arguments are readily comprehensible and thoroughly analyzed.

Response :  We would like to express our gratitude to the reviewer for your valuable time and efforts you have spent in the review. Thank you very much!

Reviewer 2 Report

This paper proposes a priority-based pre-matching strategy and also improves the PrefixSpan by considering 20 timestamps to tolerate short-term order ambiguity in alarm flood sequences. The paper is well-written and the contribution is sufficient. I would like to suggest the authors improve the paper's English writing and add some references in the years 2022 and 2023 to show your research is up to trend.  

Author Response

Response to Reviewer 2 Comments

Dear reviewer:

Thank you for your decision and constructive comments on my manuscript. We have carefully considered the suggestion of Reviewer and make some changes. We have tried our best to improve and made changes in the manuscript.

Reviewer #2:

This paper proposes a priority-based pre-matching strategy and also improves the PrefixSpan by considering 20 timestamps to tolerate short-term order ambiguity in alarm flood sequences. The paper is well-written and the contribution is sufficient. I would like to suggest the authors improve the paper's English writing and add some references in the years 2022 and 2023 to show your research is up to trend.  

Response :  Thank you for suggestions. Firstly, we would like to express our gratitude to the reviewer for your valuable time and efforts you have spent in the review. Secondly, we have carefully revised the paper by a native speaker to improve the grammar and readability. Finally, we haved added some references in the years 2022 and 2023. The added references are listed as follows:

Wang, Z.; Hu, W, Cao, W.; Wu, M. Detection of Sequential Alarm Patterns in Complex Industrial Facilities Using ClaSP and Top-K Algorithms. In 40th Chinese Control Conference, Shanghai, China, 26, July, 2021.

Alinezhad, H, S.; Shang, J.; Chen, T. Early Classification of Industrial Alarm Floods Based on Semisupervised Learning,  IEEE Transactions on Industrial Informatics, 2022, 18, 1845-1853.

Reviewer 3 Report

The results are good and satisfactory and the paper seems to be scientifically correct. The paper's argument built on an appropriate base of theory, concept or other ideas. Moreover, the research or equivalent intellectual work on which the paper is based been well designed. The abstract clearly and accurately describes the content of the article and the interpretations and conclusions have been justified by the results. The paper does demonstrate an adequate understanding of the relevant literature in the field and cite an appropriate range of literature sources. However, i want to point out the following :

1-      The paper contains some typo and grammatical errors that should be corrected. 

2-         Conclusion part of the manuscript can be developed and more explanation about the results.

3-     Finally, please check the entire manuscript carefully about the above mentioned suggestions. Last of all, this manuscript does have significant content. The topic content of this paper is consistent with the goals of the journal. So my recommendation is acceptance of the manuscript after the minor changes. It will hope fully add value to the existing literature.

Author Response

Response to Reviewer 3 Comments

Dear reviewer:

Thank you for your decision and constructive comments on my manuscript. We have carefully considered the suggestion of Reviewer and make some changes. We have tried our best to improve and made changes in the manuscript.

Reviewer #3:

The results are good and satisfactory and the paper seems to be scientifically correct. The paper's argument built on an appropriate base of theory, concept or other ideas. Moreover, the research or equivalent intellectual work on which the paper is based been well designed. The abstract clearly and accurately describes the content of the article and the interpretations and conclusions have been justified by the results. The paper does demonstrate an adequate understanding of the relevant literature in the field and cite an appropriate range of literature sources. However, i want to point out the following :

Point 1:  The paper contains some typo and grammatical errors that should be corrected. 

Response 1: Thank you very much for your suggestions. We have carefully checked our manuscript and corrected the typo and grammatical errors.

Point 2:   Conclusion part of the manuscript can be developed and more explanation about the results.

Response 2: Thank you for your suggestions. Please refer to the lines 477 to 486 on the page 16.

Point 3:   Finally, please check the entire manuscript carefully about the above mentioned suggestions. Last of all, this manuscript does have significant content. The topic content of this paper is consistent with the goals of the journal. So my recommendation is acceptance of the manuscript after the minor changes. It will hope fully add value to the existing literature.

Response 3:  Thank your for your decision and constrcutive suggestions. We have revised the mauscript according to your suggestions. Finially, we would like to express our gratitude to the reviewer for your valuable time and efforts you have spent in the review.
